# Impact of the Cooperative Health Insurance System in Saudi Arabia on Universal Health Coverage—A Systematic Literature Review

**DOI:** 10.3390/healthcare13010060

**Published:** 2025-01-01

**Authors:** Ahmed Ali Alzahrani, Milena Pavlova, Nizar Alsubahi, Ala’eddin Ahmad, Wim Groot

**Affiliations:** 1Department of Health Service and Hospital Administration, Faculty of Economics and Administration, King Abdul Aziz University, Jeddah 21589, Saudi Arabia; 2Department of Health Services Research, Care and Public Health Research Institute—CAPHRI, Maastricht University Medical Center, Faculty of Health, Medicine and Life Sciences, Maastricht University, P.O. Box 616, 6200 MD Maastricht, The Netherlands; m.pavlova@maastrichtuniversity.nl (M.P.); w.groot@maastrichtuniversity.nl (W.G.); 3Department of Marketing, School of Business, The University of Jordan, Amman 11942, Jordan; al_ahmad@ju.edu.jo; 4Maastricht Economic and Social Research Institute on Innovation and Technology, United Nations University, 6211 LK Maastricht, The Netherlands

**Keywords:** cooperative health insurance, universal health coverage, healthcare access, Saudi Arabia, health insurance impact, financial protection

## Abstract

Background: This systematic review assesses the role of the Cooperative Health Insurance System (CHIS) in achieving Universal Health Coverage (UHC) in Saudi Arabia’s evolving healthcare system by consolidating and analyzing findings from diverse studies to provide a comprehensive overview of CHIS’s impact and also identifies contextual challenges and practical insights that can inform similar reforms globally. Methods: We report results following the Preferred Reporting Items for Systematic Reviews and Meta-Analyses (PRISMA) guidelines. The following six databases were searched for relevant studies: PubMed, Scopus, CINAHL, Business Source Complete, APA PsycINFO, and SocIndex. The review protocol was registered with PROSPERO. Inclusion criteria focused on studies examining the impact of CHIS on the UHC dimensions based on the following themes: population covered, affordability, quality, efficiency, access, services covered, and financial coverage. The initial search identified 1316 publications. Results: A total of 30 studies met the inclusion criteria. Our synthesis indicates that CHIS has significantly improved healthcare access and quality, particularly in the private sector. CHIS was also associated with increased healthcare efficiency through standardized benefit packages and reduced out-of-pocket expenditures. However, these studies noted challenges such as rising insurance premiums, infrastructural deficiencies, and cultural barriers. Conclusions: CHIS is integral to Saudi Arabia’s healthcare reform, substantially contributing to UHC’s objectives. Despite notable advances, continuous efforts are needed to address existing challenges and expand coverage. The findings suggest that enhanced government support and public awareness are crucial for advancing UHC goals in Saudi Arabia.

## 1. Introduction

In Saudi Arabia, the government has long shaped the healthcare landscape, operating a mixed system with both public and private providers, where the Ministry of Health (MOH) plays the primary role in funding and regulation [1]. Furthermore, Saudi Arabia has private health insurance providers primarily serving foreign citizens, constituting a significant portion of the population [2]. The public healthcare system has undergone significant growth and a notable cost surge, placing a considerable economic burden on the nation. This burden is partly attributed to the large expatriate population comprising approximately 80% of the private sector workforce and 56% of Saudi Arabia’s total workforce. The high number of expatriates intensifies the pressure on public healthcare resources, significantly increasing the demand for medical services and thus contributing to rising costs [3].

In response to these rising costs and pressures, the government introduced the Cooperative Health Insurance System (CHIS) in 1999, a significant milestone in the country’s healthcare reform. At first, it mandated Compulsory Employment-Based Health Insurance (CEBHI) for all expatriates in the private sector, and later, that mandate was extended to all employees in the private sector and their dependents. The government also created the Council of Cooperative Health Insurance (CCHI) to supervise and regulate CHIS [4]. This initiative also represents a crucial step towards the country’s Universal Health Coverage (UHC) objectives, which are aligned with global efforts to ensure equitable access to healthcare services. CHIS is intended to mitigate part of the burden on the public healthcare system by redirecting the private sector employees and their dependents to access essential services through private healthcare providers and ensuring their financial protection against healthcare costs by making it employers’ responsibility to pay premiums entirely [5].

Currently, there is no comprehensive assessment of the impacts of the reforms, and this study addresses this gap. Therefore, the primary research question guiding this review is: how has CHIS impacted the progress toward UHC in Saudi Arabia? To answer this question, we systematically review the evidence on the direct and indirect impacts of CHIS on UHC in Saudi Arabia. Specifically, the review aims to provide insights into the effects of CHIS on achieving the fundamental principles of UHC in the Saudi healthcare system. This study is motivated by the need to understand how insurance-based models can contribute to UHC, especially in countries with mixed public–private healthcare systems. We employed the UHC definition and dimensions of the cube diagram featured in the 2010 World Health Report. In essence, UHC requires ensuring access to a comprehensive range of quality health services for all individuals without encountering financial hardship, precisely when and where they need them [6].

The added value of this systematic review lies in its comprehensive synthesis of existing research, providing a clearer understanding of the impact of health insurance on UHC in Saudi Arabia. It helps identify effective policies, highlights gaps, and offers evidence-based insights for refining the health insurance scheme towards achieving UHC. The review informs policy-making and strategic planning, guiding efforts to improve access, equity, and efficiency within the Saudi healthcare system. In essence, it provides stakeholders with knowledge to make informed decisions that support the progression toward UHC, benefiting the entire population. By addressing specific gaps in CHIS, this review may also be useful for other healthcare systems because it provides a robust evidence base that can inform similar health reforms in different contexts. By learning from Saudi Arabia’s experiences, other countries can apply these insights to improve their health insurance schemes and progress toward UHC. This cross-learning helps enhance global health outcomes and supports the broader goal of achieving UHC worldwide.

## 2. Methodology

The study used the Preferred Reporting Items for Systematic Reviews and Meta-Analysis (PRISMA) guidelines to report the review results (see Appendix A). Data collection and article review were conducted between November 2022 and January 2024. All authors jointly developed and approved the review protocol, registered in the International Prospective Register of Systematic Reviews (PROSPERO) with the ID CRD42023373838 (CRD: Centre for Reviews and Dissemination).

### 2.1. Databases and Search Terms

The study collected data from six major databases/engines: PubMed, Scopus, CINAHL, Business Source Complete, APA Psyclnfo (EBSCO), and Soc Index. The exact search phrases used were the following: ((health insurance) OR (health care insurance) OR (healthcare insurance) OR (health coverage) OR (health care coverage) OR (healthcare coverage) OR (medical insurance) OR (medical coverage) OR (cooperative health insurance) OR (cooperative health insurance system)) AND ((KSA) OR (Saudi) OR (Kingdom of Saudi Arabia) OR (Saudi Arabia)).

### 2.2. Inclusion and Exclusion Criteria

The study inclusion and exclusion criteria were based on a population, intervention, comparison, outcome, and publication framework, as detailed in Table 1.

### 2.3. Study Selection

The study employed EndNote X20, a bibliographic reference management software, to archive titles and abstracts from the initial search. Following the systematic search, duplicate entries were removed. Two authors (AA and NA) independently evaluated articles for potential eligibility based on their titles and abstracts, marking the first screening stage. Subsequently, a second screening stage was conducted, involving a thorough assessment of the full texts against the inclusion and exclusion criteria to refine the search. The same two authors independently reviewed the full texts of potentially eligible articles for relevance. Finally, the reference lists of articles selected based on full texts were screened in the last stage of the process.

### 2.4. Assessment of Quality of Studies

To assess the quality of the included studies, we used the Critical Appraisal Skills Program (CASP) checklist, which is widely used in similar research. A methodological quality assessment framework was employed to evaluate the quality of studies using the CASP Qualitative tool and an adapted version of the CASP tool for Quantitative studies [7]. Studies were rated based on these tools, and the overall quality rating percentage was calculated. The studies were then categorized as excellent (85–100%), good (66–84%), fair (34–65%), and low quality (0–33%). In total, 14 studies were rated excellent and 9 good, while 7 were rated fair (Figure 1). Appendix A (in Appendix A) show the complete quality assessment results for the included studies.

### 2.5. Data Extraction and Analysis

Two independent reviewers (AAA and NA) initially extracted the data using a standardized Excel-based extraction sheet (see Appendix A). The extracted data encompassed authors’ names, publication year, study aim, design, population, and sample details. We also extracted and summarized the main findings reported in the publications reviewed. It also included information on the type of data collected, study location, methods employed, any limitations identified, practical implications, and suggestions for future research. Findings were subsequently verified by a third and fourth reviewer (M.P. and W.G.).

### 2.6. Data Synthesis and Reporting

The study utilized a directed qualitative content analysis method. This method offers a structured framework for identifying themes for analysis, facilitating systematic data exploration. Relying on the established concept of the UHC cube outlined by the WHO for data extraction, we synthesized the following themes: population covered, affordability, quality, efficiency, access, services covered, and financial coverage. Specifically, our review focused on these seven themes delineated based on the UHC concept to extract, analyze, and present data concerning the impact of CHIS on UHC objectives in Saudi Arabia.

## 3. Results

### 3.1. Search Results

The initial search identified 1316 publications, from which 1175 unique records remained after duplicate removal. Of these, 39 were deemed potentially eligible based on the screening of titles and abstracts and subsequently underwent full-text review. A total of 1136 records were excluded for specific reasons: a significant portion did not directly address CHIS or its impact on UHC dimensions in Saudi Arabia, failing to meet the relevance criteria. Additionally, studies focused on populations outside Saudi Arabia, those lacking empirical outcomes related to UHC dimensions, or non-research publications such as opinion pieces and Editorials were excluded. After full-text screening, 30 studies met the inclusion criteria and were analyzed (Figure 2).

### 3.2. General Description of the Selected Articles

The majority of publications included in the analysis, 43.3%, were published between 2020 and 2024, followed by 33.3% between 2015 and 2019. In terms of the location of the study, 50% of the articles were conducted at the country level, while 33.3% focused on Riyadh city, 10% on Jeddah city, and 6.6% on the regional level. Regarding the aim of the articles, 63.3% aimed to assess, 26.7% aimed to describe, and 10% aimed to propose reform in the context of health insurance and UHC in Saudi Arabia.

All articles included in the analysis were research papers, as this was an inclusion criterion, with 76.6% adopting a quantitative research approach and 23.3% utilizing a qualitative approach. Regarding data collection/design, 53.3% of the articles utilized questionnaires, 33.3% relied on secondary data, and smaller percentages employed informant interview surveys (3.3%), structured interviews (3.3%), and semi-structured interviews (6.7%).

### 3.3. Summary of the Characteristics of the Studies Included

#### 3.3.1. Population Covered

Seventeen studies reported on the population covered (see Table 2). Baseline data on the insured population before CHIS implementation are not clearly established in the reviewed papers, likely due to limited publicly available data and reliance on the same government sources. Generally, the health insurance market was minimal before CHIS, limited mainly to some expatriates and wealthier Saudis who wanted access to private healthcare services, while most of the population relied on public healthcare. Studies [8,9,10,11] described the phased approach to CHIS enrollment alongside the existing public healthcare. This approach began with the enrollment of expatriates and their dependents, followed by Saudis working in the private sector and their dependents, and ultimately aimed to enroll the entire population [12,13,14,15,16,17]. These phased strategies aimed to ensure comprehensive coverage across the population over time. Because of this plan, 8.4 million individuals were enrolled in 2010, of whom 6.47 million were expatriates and 1.87 million were Saudis [18]. By 2016, the plan covered 38% of the population, with a predominant share being expatriates, totaling more than 12 million [14,19]. In 2020, the population enrolled in CHIS decreased to 10.5 million due to a change in employment patterns and regulations for expatriates, leading many of them to leave the country [20].

While progress has been made in expanding enrollment, challenges persist, as [21] noted, resulting in only the first two phases being successfully implemented (coverage for expatriates and their dependents, followed by Saudis working in the private sector and their dependents). However, refs. [22,23] indicated ongoing government evaluations regarding the feasibility of completing the third phase (aimed to enroll the entire population), reflecting a commitment to expanding coverage and access to healthcare services.

#### 3.3.2. Affordability

Ten studies reported on affordability (see Table 2). The cost of healthcare is a decisive factor for patients in Saudi Arabia, regardless of insurance status [24]. Concerns about affordability under CHIS were raised, particularly for low-income expatriates, as it is difficult for them to pay OOP for copays or services that their plan does not cover [17]. CHIS policies include affordability measures designed to ease those concerns, such as predetermined maximum copayments (not to exceed 20% of healthcare expenses or a maximum of USD 26.67 per treatment), no deductibles for inpatient services and prescription drugs, and a USD 533.30 annual cap for dental treatment [25]. However, households enrolled in CHIS exhibited higher overall OOP health expenditure than those who were not [26]. In addition, some individuals are forced to bear OOP expenses for essential care due to limited coverage plans or overcrowded public healthcare facilities [12].

Varying willingness to pay to participate in any future national health insurance schemes was revealed, influenced mainly by household income levels [9,14,27]. This disparity highlights the disproportionate financial burden of insurance premiums on lower-income individuals. In addition, insurance premiums in CHIS are linked to affordability risk as they increase by approximately 15% on average annually [28]. This increase in premiums was needed to minimize the financial shortages [21]. However, premium increases also pose challenges for employers striving to balance effective employee healthcare coverage, as required by law, with employers’ cost containment and operational expenses.

#### 3.3.3. Quality

Seven studies reported on quality (see Table 2). Private enrollment into CHIS appeared to introduce selection bias, as individuals valuing quality prefer treatment in private hospitals [24]. Additionally, Saudis not currently enrolled in CHIS expressed a willingness to pay to enroll in a possible national health insurance scheme to access this perceived better quality care [16]. The availability of CHIS services and the perceived service quality were positively related to expatriates’ satisfaction [28]. Disparities in service quality between insurance classes are evident. Namely, higher-class plans, such as Class A, VIP, or Gold, offer a more extensive network of healthcare facilities, thereby ensuring higher service quality. These classes, which vary by insurance company, generally progress from basic (or Class C) to the premium tiers, with each tier offering progressively better quality services and healthcare options [17,20].

Most research on the quality of care was conducted in MOH facilities and relatively few in the private sector; thus, a complete assessment of the effect of insurance is not possible, and efforts to enhance the overall quality of care remain fragmented and uncoordinated [18]. Lack of awareness, knowledge, and understanding of health insurance options and rules and regulations related to CHIS plans also undermines the overall quality and abundance of care received [29].

#### 3.3.4. Efficiency

Ten studies reported on efficiency (see Table 2). Before CEBHI was implemented, medical coverage exhibited significant variability in coverage extent, provider networks, and benefit limitations. CCHI now plays a central role in standardizing benefit packages and overseeing insurance companies, healthcare providers, and financial regulations, reducing these variations and monitoring compliance [9,13,21]. The review findings show that challenges impeding the full implementation of the CHIS Act include weak infrastructures, escalating insurance premiums, claim rejections, and insurance fraud [18,30].

Additionally, more generous insurance plans are often prone to moral hazard, resulting in unnecessary medical treatments or opting for more costly services than necessary due to the reduced personal cost burden. This phenomenon arises because the direct financial impact of healthcare decisions is mitigated by insurance coverage, leading to an overutilization of healthcare resources [26]. The recommendation is to organize services based on patients’ profiles and implement effective referral systems across all levels of care to bolster efficiency [31].

Despite these challenges, the implementation of CEBHI has yielded positive outcomes, especially in healthcare provider payment methods, leading to lower OOP payments and increased contributions from private health insurance [20]. Moreover, timely approval processes for treatments and related services were highlighted, with 87.5% of requests approved within minutes, meeting the 60 minutes requirement set by CCHI [32]. However, some inefficiencies persist among insurance companies, such as delays in claims payments and the absence of electronic transfer protocols and key performance indicators [33].

#### 3.3.5. Access

In total, 18 studies reported on access (see Table 2). After the introduction of CEBHI, expatriates were directed toward private healthcare facilities, and now they have no access to public healthcare facilities [12,27]. MOH data reveal a consistent improvement in access over time due to increased inpatient and outpatient services offered within the private healthcare sector. This improvement can be attributed to a growing number of workers in the private sector enrolled in CHIS, which reduces financial barriers to healthcare and encourages more frequent use of medical services. Additionally, there is a preference for privately purchased subscriptions to CHIS among high-income groups [20]. However, private insurance does not change service utilization compared to direct public healthcare access [34]. One study noted limitations in privately purchased subscriptions due to social beliefs, such as the belief that insurance is against Islamic law [22]. In addition, findings from the National Family Health Survey indicated that CHIS subscribers, particularly expatriates, demonstrated a higher inclination toward health check-ups [20]. A study by [8] found a positive correlation between enrollment status and healthcare service utilization. This correlation was higher among individuals who privately purchased subscriptions than those covered by employers [3] and individuals with chronic conditions [26]. Conversely, one study found that insured individuals used Primary Health Centers (PHCs) less than uninsured individuals, suggesting that the availability of insurance might shift patient preferences toward more specialized care facilities, even for basic healthcare needs [35]. Married expatriate workers were found to have better coverage due to their partners’ income [13]. Additionally, expatriates’ knowledge of insurance benefits was negatively associated with a lack of access to healthcare services [19].

The literature reviewed suggests that urban–rural imbalances and regional disparities are persistent, causing delays in delivering insurance-covered services across the population. These disparities mean that individuals in urban areas generally have quicker access to healthcare services due to the higher concentration of healthcare facilities and professionals, in contrast to rural areas [21,36]. Prolonged approval processes from insurance companies and limited patient choices also contribute to these disparities, as some physicians prefer to treat only cash-paying patients [14,31]. Denials of subscription to plans and refusals by insurance companies to authorize payment for treatments and related services are uncommon and typically resolved through negotiation with CCHI rather than arbitration or judicial action [33]. Individuals enrolled in CHIS were found to enjoy more convenient access due to improved administrative procedures [24]. However, despite having CHIS coverage, access to advanced care remained contingent on income [15]. Higher-income earners were found to have better access to advanced care through either superior insurance plans or their ability to pay OOP, a flexibility unavailable to lower-income groups.

#### 3.3.6. Services Covered

Nine studies provided insights into the services covered (see Table 2). Refs. [8,15] emphasized the comprehensive nature of CHIS policies, covering essential medical services, consultations, follow-up visits, and referrals. However, there are limits on coverage for some services. For example, CHIS policies impose a maximum coverage of expenses for dental care, renal dialysis, optical services, and medications. Insurance companies typically do not limit the number of drugs per prescription but may require pre-authorization for costly medications [32].

CCHI oversees unified health insurance benefit packages, ensuring coverage for necessary medical examinations, treatments, medications, and diagnostic procedures [3,13,25]. This comprehensive coverage aims to include a wide range of essential health services. However, there are limitations in the coverage for services outside the defined package network, which may not be approved. Additionally, Refs. [12,17,20] identified discrepancies in insurance packages regarding expenditure caps and provider networks. This finding means that despite the intended comprehensive service coverage in CHIS, variations in insurance plans can lead to inconsistencies in the availability of certain health services, thus impacting the overall goal of achieving uniform service coverage across all insured individuals.

#### 3.3.7. Financial Coverage

Twelve studies provided insights into financial coverage (see Table 2). Ref. [20] revealed the transition from OOP spending and voluntary employer-provided health insurance to mandatory coverage under the CEBHI scheme. Employers in the private sector now contribute the total premium, with penalties imposed for non-compliance [8,18]. Implementing CEBHI has reduced OOP payments, indicating improved financial coverage [17,31]. The CEBHI scheme not only requires employers to bear the total cost of enrollment into CHIS but also determines unified benefit packages managed by the CCHI [19,25,27,32]. The CEBHI scheme offers protection against rising premium costs over time and specifies a maximum on accumulated copayments for all employees and their dependents [13]. However, insurance classes that require additional copayments for medicines might indicate potential gaps in financial coverage due to high accumulated OOP payments [15,36].

## 4. Discussion

CHIS markedly shapes healthcare services in Saudi Arabia, particularly in private hospitals, where it enhances access for a broad spectrum of the population. The facilitation of increased healthcare utilization contributes to a more convenient healthcare experience, clearly demonstrating the pivotal role of health insurance schemes in the broader healthcare landscape [27]. The findings of this review have significant practical implications for insurance providers and beneficiaries. For providers, the adoption of standardized benefit packages and monitoring frameworks, as observed under CHIS, highlights opportunities to enhance service efficiency while minimizing out-of-pocket costs. Providers can also address identified gaps, such as disparities in rural access, by investing in infrastructure and tailoring premiums to income levels. For beneficiaries, the expansion of CHIS coverage has the potential to alleviate financial burdens and improve access to essential services, particularly for low-income groups. Policymakers and providers must ensure that reforms prioritize equitable access across socioeconomic and geographic divides to maximize the benefits of CHIS for all stakeholders. The system currently grapples with significant challenges, including cultural perceptions such as the permissibility of insurance in Islam, rural–urban imbalances, and limited public awareness. These challenges necessitate targeted interventions by CHIS to expand access and overcome those barriers to ensure equitable distribution of healthcare resources [21,36].

The quality of healthcare services remains a critical area of scrutiny. Patient preferences for private hospitals often reflect a perceived higher quality of services, which might be compromised as insurance companies and hospitals may prioritize cost reductions [24,28]. Third-party payers are thus vital in maintaining high standards to ensure quality is not sacrificed for cost efficiency.

Merits of efficiency within the health insurance system are present, but obstacles such as infrastructural issues, rising premiums, and insurance fraud might hinder its full optimization. CEBHI has positively impacted payment methods, although concerns about rising premiums for high-risk groups underline the need for a balanced approach in health insurance management [9,18,30]. Regulatory frameworks must evolve to address these issues effectively and balance risk and premiums fairly.

Financial considerations heavily influence patient choices, with a noted preference for private hospitals regardless of insurance enrollment status [24]. The financial implications of healthcare premiums, especially for lower-income individuals, are a recognized concern that impacts participation rates in any future national health insurance scheme [21]. Exploring income-sensitive insurance schemes is essential to ensuring widespread participation and maintaining affordability. While the reviewed studies suggested CHIS has reduced out-of-pocket costs, none specifically discuss catastrophic health spending before or after its implementation, leaving a gap in understanding the full financial impact on households.

Saudi Arabia’s healthcare system has significantly expanded coverage through the phased implementation of CEBHI, initially covering expatriates and gradually including Saudi nationals and their dependents [18,20,21]. Despite this successful expansion, the need for continuous efforts to achieve UHC remains a priority, especially with the government’s intention to privatize state-owned hospitals, emphasizing this goal.

The uniformity of benefits across various health services under CHIS plays a crucial role in this expansion. However, addressing variations in access levels based on insurance categories is vital for achieving comprehensive and equitable healthcare services for all beneficiaries [8,14].

### 4.1. Limitations

The reviewed studies underscore various limitations that merit careful consideration. The geographic concentration of studies within a specific city or region in Saudi Arabia limits the generalizability of their findings to the broader population. Furthermore, few studies specifically address rural areas, where disparities in healthcare access may be more pronounced, or provide detailed insights into demographic subgroups such as low-income populations, women, or children. This lack of representation limits the ability to generalize findings to these populations, underscoring the need for future research focusing on these groups. Sampling issues emerged as a recurring challenge across multiple studies, as the reliance on convenience sampling introduces potential biases and limits the representativeness of the study populations. Consequently, these geographic and demographic limitations may skew findings, potentially underrepresenting healthcare challenges and disparities in rural areas. Specific demographic groups, such as ages below 18 and the female gender, were often underrepresented, affecting the overall external validity of the research. The use of cross-sectional designs and reliance on self-reported data make it impossible to establish causal relationships and ensure data accuracy. This constraint implies that observed associations between CHIS coverage and UHC outcomes should be interpreted cautiously, acknowledging the need for causal validation in future research. The identified common method bias further complicates the interpretation of results.

### 4.2. Implications for Policy and Research

This review underscores the need for policymakers to enhance private health insurance with robust government support and to focus on increasing public awareness about health insurance and UHC. Future research should extend beyond current methodologies, incorporating larger, more diverse samples from public and private hospitals to ensure comprehensive data comparison. Specific policy adjustments, such as income-sensitive premiums and regional service enhancements, could address identified disparities in access and affordability. More interdisciplinary research is needed to examine the effects of health policies on medication access and the broader implications of the CEBHI scheme, particularly regarding its impact on expatriates and copayment structures. Future research should also utilize longitudinal studies across various regions to capture the dynamics of health insurance impact over time and ensure that findings represent the entire country. Studies should explore the practical challenges of current insurance frameworks and consider the perspectives of insurance providers and recipients.

## 5. Conclusions

This systematic review sheds light on how CHIS influences the achievement of UHC in Saudi Arabia. Examining various dimensions, it emphasizes the intricate relationship between health insurance and the pursuit of UHC. The findings underscore the importance of targeted reforms to alleviate financial barriers and improve access for the populations. Additionally, enhancing healthcare accessibility in rural regions, where disparities are more pronounced, could significantly advance UHC goals. For healthcare providers, understanding the nuances of CHIS benefits across different insurance classes may help guide patients more effectively, ensuring they receive appropriate care within their coverage limitations. The findings highlight actionable areas for policymakers, such as addressing premium affordability and enhancing rural healthcare access, which could significantly advance UHC goals. These insights are crucial for healthcare providers and policymakers in Saudi Arabia, offering guidance to advance UHC goals. Saudi Arabia could make substantial strides toward comprehensive and inclusive UHC by implementing region-specific policies and income-sensitive insurance models. The identified gaps point to opportunities for future research and highlight the need for ongoing efforts to improve the healthcare system. This review underscores the importance of continuous research within Saudi Arabia to deepen our understanding of its healthcare system and inform practical policy reforms for equitable healthcare coverage.

## Figures and Tables

**Figure 1 healthcare-13-00060-f001:**
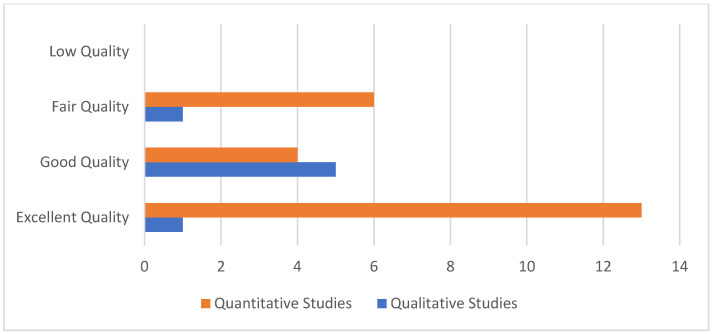
Quality assessment of included studies.

**Figure 2 healthcare-13-00060-f002:**
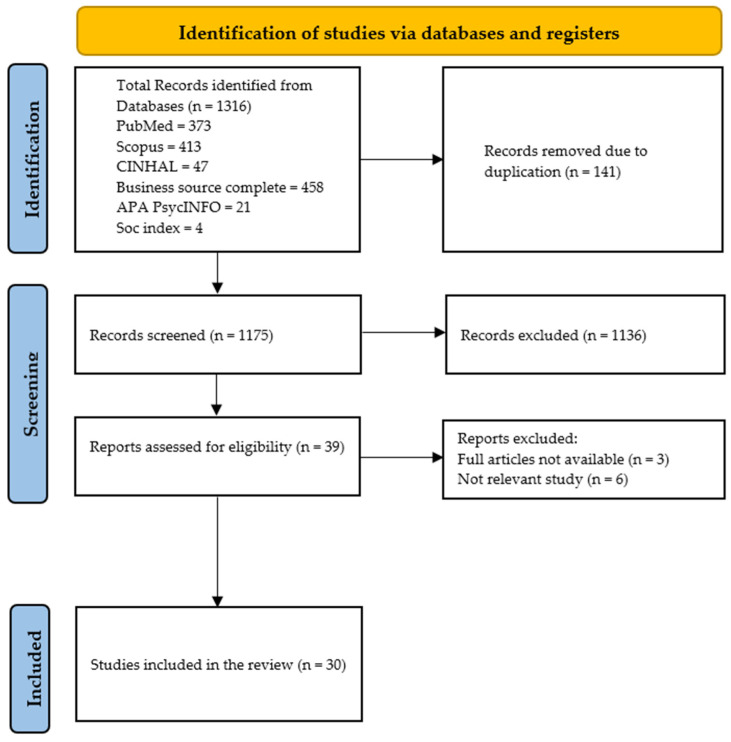
PRISMA flowchart.

**Table 1 healthcare-13-00060-t001:** Inclusion and exclusion criteria.

Criteria	Inclusion Criteria	Exclusion Criteria
Population	Kingdom of Saudi Arabia	Not the Kingdom of Saudi Arabia
Intervention	Health insurance	Not health insurance
Comparison	Relation between health insurance and universal health coverage	No relation to health insurance and/or universal health coverage
Outcome	Universal health coverage definition/dimensions: access, quality, efficiency, affordability, population covered, service covered, financial coverage	Other dimensions not related to universal health coverage
Publication	All published articles in English and Arabic that report qualitative, quantitative, or mixed-method study results	Duplicate publications or publications not reporting study results

**Table 2 healthcare-13-00060-t002:** General characteristics of publications included in the analysis (N = 30 publications reviewed).

Classification Category	Subcategories	N (%) *	Reference Index in Appendix A
Year of publication	Before 2004	1 (3.3%)	(1)
	2005–2009	1 (3.3%)	(2)
	2010–2014	5 (16.6%)	(3, 4, 5, 6, 7)
	2015–2019	10 (33.3%)	(8, 9, 10, 11, 12, 13, 14, 15, 16, 17)
	2020–2024	13 (43.3%)	(18, 19, 20, 21, 22, 23, 24, 25, 26, 27, 28, 29, 30)
Location of study	Country level	16 (53.3%)	(2, 4, 5, 6, 7, 9, 11, 17, 19, 21, 22, 23, 24, 26, 29, 30)
	Regional level	2 (6.6%)	(8, 27)
	Riyadh city	9 (30%)	(1, 10, 12, 13, 14, 18, 20, 25, 28)
	Jeddah city	3 (10.0%)	(3, 15, 16)
Aim	Assess	20 (66.6%)	(2, 3, 6, 7, 9, 11, 12, 15, 17, 19, 20, 21, 22, 23, 24, 25, 26, 27, 28, 30)
	Propose reform	2 (6.6%)	(8, 18)
	Describe	8 (26.6%)	(1, 4, 5, 10, 13, 14, 16, 29)
Type of publication	Research paper	30 (100%)	(1, 2, 3, 4, 5, 6, 7, 8, 9, 10, 11, 12, 13, 14, 15, 16, 17, 18, 19, 20, 21, 22, 23, 24, 25, 26, 27, 28, 29, 30)
Research approach	Qualitative	7 (23.3%)	(4, 5, 15, 17, 21, 26, 30)
	Quantitative	23 (76.6%)	(1, 2, 3, 6, 7, 8, 9, 10, 11, 12, 13, 14, 16, 18, 19, 20, 22, 23, 24, 25, 27, 28, 29)
Data collection/design	Questionnaires	17 (56.6%)	(1, 2, 3, 4, 9, 10, 11, 12, 13, 14, 15, 17, 19, 21, 28, 29, 30)
	Informant interview survey	1 (3.3%)	(7)
	Structured interview	1 (3.3%)	(26)
	Semi-structured interview	2 (6.6%)	(16, 22)
	Secondary data	9 (30%)	(5, 6, 8, 18, 20, 23, 24, 25, 27)
Theme	Population covered	17 (56.6%)	(2, 3, 4, 7, 12, 14, 17, 19, 20, 22, 23, 24, 25, 26, 27, 29, 30)
	Affordability	10 (33.3%)	(1, 7, 9, 13, 16, 17, 19, 24, 26, 28)
	Quality	7 (23.3%)	(1, 4, 7, 9, 11, 29, 30)
	Efficiency	10 (33.3.0%)	(4, 5, 6, 8, 12, 17, 21, 26, 28, 30)
	Access	18 (60.0%)	(1, 2, 3, 5, 8, 10, 12, 14, 15, 16, 17, 18, 19, 22, 23, 24, 28, 30)
	Services covered	9 (30.0%)	(2, 6, 7, 12, 13, 22, 23, 24, 30)
	Financial coverage	12 (40.0%)	(2, 4, 5, 6, 7, 12, 13, 14, 15, 16, 23, 30)

* Note that the total sum per category might exceed N because the same paper might be classified into more than one category.

## Data Availability

Data relevant to the study are either included in the article or uploaded as additional files.

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
