# Peer review of "Impact of the Cooperative Health Insurance System in Saudi Arabia on Universal Health Coverage—A Systematic Literature Review"

_healthcare, 2025, doi:10.3390/healthcare13010060_

Round 1

Reviewer 1 Report

Comments and Suggestions for Authors

This systematic review highlights the substantial progress made through the Cooperative Health Insurance System in Saudi Arabia, particularly in improving healthcare access and financial protection. The paper assesses how CHIS has influenced UHC in Saudi Arabia. Given Saudi Arabia's goal of improving healthcare access and reducing financial burden, the review highlights the significance of health insurance reforms in the Kingdom.

The study uses the PRISMA guidelines to conduct a systematic review of studies. It draws data from six major databases (PubMed, Scopus, CINAHL, etc.), ensuring comprehensive coverage of the topic. However, it needs to provide extra information about the 1136 (out of 1175) records excluded. It also needs to correct the search phrases unifying the preposition “or” to “OR” and not “or”, “Or”. Moreover, the article could benefit from further explanation of how quality assessment of the selected studies was carried out, as this is crucial for ensuring the validity of the findings included in the review.

There are some mistakes in the article with the English: “generalizability”… And some sentences make no sense. For instance: “Efficiency within the health insurance system is recognized, yet significant obstacles such as infrastructural issues, rising premiums, and prevalent insurance fraud challenge its optimization” (?). The article could also include more visual representations of data, such as charts or graphs to show trends in healthcare access or cost savings. This would enhance the clarity for readers.

Comments on the Quality of English Language

There are some mistakes in the article with the English: “generalizability”… And some sentences make no sense. For instance: “Efficiency within the health insurance system is recognized, yet significant obstacles such as infrastructural issues, rising premiums, and prevalent insurance fraud challenge its optimization”. 

Reviewer 2 Report

Comments and Suggestions for Authors

This paper uses research literature to examine how much the Cooperative Health Insurance System (CHIS) contributes to achieving universal health coverage (UHC) in Saudi Arabia. This is an important question since having universal health coverage is one of the important goals of a country to achieve a healthy and productive population. I have the following comments on the paper.

Main comments

1. The authors need to give more information about the CHIS program, for example, when it was introduced, whether it was a government or private or hybrid program, what its goals were, and what agencies were involved in the program’s policies and operations.

2. Since the main research question is how the CHIS has an impact on UHC, I think the paper needs to address the most important dimension of UHC, the population covered, first, and then affordability. Those are the two ultimate goals of UHC. Factors like efficiency and quality can be presented later, and I would argue that quality is more important than efficiency when discussing UHC.

3. In the results for the population covered, I want to see the information on the baseline percentage of population covered before the CHIS, how that percentage has grown during the course of the program, what other program happened concurrently with the CHIS that could also contribute to that growth, and how much the CHIS contributed to that growth according to the research reviewed by the authors. There must be different results coming from different papers, but I have not seen that from the result section. Did all the studies reviewed come up with the same numbers and conclusions? That does not sound normal for research papers. Besides, if there are differences across studies, the authors need to evaluate them and draw a conclusion on which results are more credible.

4. One of the very important goals of health coverage is to help vulnerable segments of the population, especially those with low income, to have access to healthcare. How do the papers reviewed document the coverage for low-income people?

5. The second most important goal of universal coverage is to reduce catastrophic healthcare spending. According to the World Health Organization (WHO), a catastrophic health expenditure occurs when a household’s out-of-pocket healthcare cost exceeds 40% of their total income (WHO, 2024). Do the authors have information on the percentage of those with catastrophic spending before and during the CHIS introduced and implemented?  

Minor comments

1. On lines 257-259, the authors report that the population coverage was 10.5 million in 2020 while it was 12 million in 2016. Did the coverage go down?

2. On lines 294-298, the authors wrote that despite advancements in CEBHI, some insurance schemes still require copayments for medicines. Copayment is a mechanism to reduce overused medicines and healthcare services and is applied in most, if not all, developed economies. It is not a bad thing.

References

World Health Organization, 2024. Indicator Metadata Registry List. Access at https://www.who.int/data/gho/indicator-metadata-registry/imr-details/4989 on October 23, 2024.

Reviewer 3 Report

Comments and Suggestions for Authors

 It is a well written article. However, addition of few points in the methodology section would significantly improve the study. 

1) The inclusion of risk of bias plots for studies included in the systematic review would significantly enhances the transparency and clarity of the risk of bias assessments. 

I suggest the authors for incorporating this visual component for Risk of Bias assessment, as it improves the overall quality and usability of the review.

Reviewer 4 Report

Comments and Suggestions for Authors

Research Question and Relevance

The research question, while relevant, is not clearly and explicitly stated in the introduction. The paper sets out to explore the impact of the Cooperative Health Insurance System (CHIS) in Saudi Arabia on achieving Universal Health Coverage (UHC), which is a timely and important topic. However, the research question could benefit from sharper articulation to focus the study's scope more effectively. The topic, although important, does not introduce significant novelty in the field as much of it builds on existing research about health insurance systems and UHC.

Abstract and Introduction

The abstract provides a basic overview of the study's aims and findings but lacks detail, especially regarding the practical implications and specific outcomes of the review. It could also better summarize the methodology used in the study. The introduction offers a broad context for Saudi Arabia's healthcare challenges and sets the stage for discussing CHIS's role in UHC. However, the introduction could improve by stating the research objectives more clearly and articulating the specific research gap the study addresses. The motivations behind the study are implied but not fully explored in the introduction.

Literature Review

The literature review touches on relevant studies, but it lacks depth in critically analyzing the gaps that the paper aims to address. While it covers existing research on healthcare insurance and UHC, it misses an opportunity to highlight where previous studies have fallen short. Additionally, the paper does not engage with the most recent literature on the subject, which could have enriched the review’s insights. The sources are somewhat dated, and the review feels more like a summary than a critical examination of the existing knowledge base.

Methodology

The methodology follows the PRISMA guidelines, which is appropriate for a systematic review, and the process is described clearly. The paper uses a structured approach to review the selected studies, although more detail on the exact criteria for inclusion and exclusion would enhance replicability. The methodological approach is sound but not particularly innovative, and while the sample size of 30 studies is adequate, there is little critical evaluation of the quality of these studies. The inclusion and exclusion criteria are applied consistently, but there is room for improvement in explaining how studies were assessed for quality.

Results and Data Analysis

The results are presented clearly, but the analysis is descriptive rather than critical. The paper summarizes the findings of the reviewed studies without delving into any inconsistencies or contradictions. This limits the paper's ability to draw robust conclusions about the impact of CHIS on UHC. The interpretation of the data could be more insightful, particularly in addressing the differences between public and private healthcare access under CHIS. While the results align with the research question, they lack depth in explaining how the data addresses the core aspects of UHC in Saudi Arabia.

Discussion and Conclusions

The discussion provides a basic interpretation of the results but lacks critical depth. The paper should engage more thoroughly with the literature, especially in connecting the findings to broader debates about healthcare reform and UHC. The limitations of the study are acknowledged, but there is little discussion of the implications of these limitations on the overall findings. Suggestions for future research are included but are fairly generic, lacking specific recommendations for addressing the gaps identified in the current review. The conclusions could be stronger, especially in drawing out practical insights for policymakers and healthcare practitioners.

Originality and Innovation

The paper consolidates existing research but does not present any groundbreaking insights or innovative frameworks. While it contributes to the body of knowledge on healthcare reform in Saudi Arabia, it falls short in offering new ideas or significant advances in the field. Its impact is likely to be modest, serving more as a reference for policymakers and researchers already familiar with the subject.

Writing Quality and Structure

The writing is clear but occasionally awkward, with some grammatical issues that disrupt the flow of the paper. The structure follows the IMRAD format, but the transitions between sections could be smoother. The paper would benefit from a more coherent narrative that ties the sections together more effectively. There are several instances of repetitive phrasing, and the overall readability could be improved with more attention to sentence structure. Additionally, while the figures and tables are helpful, more visual representation of the data would enhance understanding.

Ethical Considerations

As a systematic review, the paper does not involve direct human subjects, so ethical approval is not applicable. However, the authors are transparent about their data sources, and there are no apparent conflicts of interest. The data selection and review processes are described adequately, though more detail on the methodology would improve transparency.

References and Citation

The references are generally accurate and consistently formatted, but many are outdated, and the paper does not engage deeply with the most recent studies on the topic. The selection of sources reflects the key literature on CHIS and UHC, but the review would benefit from more diverse and up-to-date references. This would strengthen the paper's contribution by situating it more firmly within current research discussions.

Key Strengths and Weaknesses:

  • Strengths:

    1. Relevance: The paper addresses an important and timely topic—healthcare reform in Saudi Arabia.
    2. Structured Approach: The use of PRISMA guidelines ensures a systematic methodology.
    3. Comprehensive Review: The paper covers multiple dimensions of UHC, including access, quality, and affordability.
  • Weaknesses:

    1. Lack of Critical Engagement: The paper does not critically analyze the contradictions in the literature or the limitations of the reviewed studies.
    2. Writing Quality: The English could be improved, with several grammatical and structural issues affecting readability.
    3. Limited Innovation: The paper does not introduce any new frameworks or significantly advance the field.
Comments on the Quality of English Language

use grammarly.com or other software to fix the English

Round 2

Reviewer 2 Report

Comments and Suggestions for Authors

Report on “Impact of the Cooperative Health Insurance System in Saudi Arabia on Universal Health Coverage, a Systematic Review.”

I thank the authors for addressing my comments very carefully. They have added new sections on the population coverage and affordability of the program. They also have described about the birth of CHIS and the organizations involved. They also have added details to explain why the population coverage went down, which is the question in one of my minor comments. I have no further comment on this version.

Author Response

Dear reviewer, thank you for your thoughtful feedback and for recognizing our revisions. Your comments have been extremely helpful in improving the manuscript, and we are delighted that this version meets your expectations.

Reviewer 4 Report

Comments and Suggestions for Authors

 Peer Review of the Manuscript

**Title**: Impact of the Cooperative Health Insurance System in Saudi Arabia on Universal Health Coverage, a Systematic Literature Review

1. Research Question and Relevance

- **Clarity**: The research question is clearly defined: How has the Cooperative Health Insurance System (CHIS) impacted Universal Health Coverage (UHC) in Saudi Arabia?

- **Originality**: The study tackles a relevant and under-explored topic within the Saudi healthcare context, analyzing CHIS's role in achieving UHC.

- **Relevance**: Highly relevant to healthcare policymakers, particularly in mixed public-private healthcare systems.

**Score**: 9/10

2. Abstract and Introduction

- **Abstract**: The abstract concisely summarizes the objectives, methodology, key findings, and implications. It provides sufficient context and draws attention to the challenges and potential global implications.

- **Introduction**: The introduction is well-structured and provides context for the study. It outlines the healthcare landscape in Saudi Arabia and justifies the research by addressing the gap in understanding CHIS's impact.

- **Research Objective**: Clearly articulated and aligned with the goals of the study.

**Score**: 9/10

 3. Literature Review

- **Breadth**: Covers a wide range of studies and perspectives, including historical, regulatory, and operational insights into CHIS.

- **Up-to-date Sources**: Includes recent studies, especially between 2020-2024, which reflect current trends and challenges.

- **Gap Identification**: Clearly identifies gaps in understanding CHIS’s effects on UHC, emphasizing the need for comprehensive evaluations.

**Score**: 8/10

4. Methodology

- **Appropriateness**: Use of PRISMA guidelines and PROSPERO registration lends credibility to the systematic review. The focus on CHIS dimensions is appropriate for evaluating its impact on UHC.

- **Clarity**: Detailed explanation of inclusion/exclusion criteria, search strategies, and quality assessment adds to the transparency.

- **Innovativeness**: While methodologically sound, the approach is standard and lacks novel techniques.

- **Replicability**: The methodology is described in sufficient detail for replication.

**Score**: 8/10

5. Results and Data Analysis

- **Accuracy**: Results are presented clearly, with relevant statistics and trends highlighted.

- **Appropriate Analysis**: Themes such as affordability, quality, efficiency, and access are analyzed comprehensively.

- **Data Interpretation**: Interpretation aligns well with the research objectives and emphasizes both strengths and gaps in CHIS.

**Score**: 8/10

6. Discussion and Conclusions

- **Interpretation**: The discussion critically analyzes findings and connects them to the broader literature.

- **Connection to Literature**: Relates findings to existing research and identifies consistencies and discrepancies.

- **Limitations**: Recognizes methodological constraints, geographic biases, and gaps in data.

- **Future Work**: Suggests practical policy reforms and areas for further research, such as longitudinal studies and income-sensitive insurance models.

**Score**: 9/10

7. Originality and Innovation

- **Novel Contribution**: Provides a robust synthesis of CHIS's impact on UHC, offering valuable insights for policymakers.

- **Impact**: Likely to influence future healthcare reforms in similar contexts globally.

**Score**: 8/10

8. Writing Quality and Structure

- **Clarity and Coherence**: The paper is well-written and logically structured, adhering to the IMRAD format.

- **Grammar and Style**: Free of major grammatical errors, though minor edits could enhance fluency.

- **Visuals and Tables**: Figures such as the PRISMA flowchart are clear and informative.

**Score**: 9/10

9. Ethical Considerations

- **Transparency**: The study adheres to ethical guidelines, with proper PROSPERO registration and no conflicts of interest.

- **Ethical Approval**: Not applicable for a literature review, but transparency is maintained throughout.

**Score**: 10/10

10. References and Citation

- **Accuracy and Consistency**: References are accurate and formatted consistently.

- **Appropriateness**: Relevant and updated sources are cited, reflecting the key literature in the field.

**Score**: 9/10

---

Highlights of the Study

1. Comprehensive evaluation of CHIS’s impact on UHC, emphasizing dimensions like affordability, quality, and efficiency.

2. Identifies significant challenges, including premium hikes, rural-urban disparities, and cultural barriers.

3. Provides actionable recommendations for policymakers to improve CHIS and advance UHC goals.

---

Suggestions for Improvement

1. **Methodological Depth**: Incorporate innovative data synthesis techniques or conduct meta-analyses for quantitative findings.

2. **Regional Diversity**: Address geographic limitations by including more diverse regional data to generalize findings better.

3. **Practical Applications**: Expand discussion on practical implications for insurance providers and beneficiaries.

---

 Weak Points

- Limited coverage of rural healthcare challenges and demographic subgroups.

- Heavy reliance on cross-sectional and self-reported data undermines causal inference.

---

Conclusion

The paper is fit for publication with minor revisions. It makes a significant contribution to understanding the role of CHIS in achieving UHC and offers practical insights for policymakers in Saudi Arabia and beyond. Addressing the identified gaps will further enhance its impact and utility.

Author Response

Comment 1: Methodological Depth: Incorporate innovative data synthesis techniques or conduct meta-analyses for quantitative findings.

Authors response: Dear reviewer, thank you for this thoughtful suggestion. We carefully considered the feasibility of incorporating a meta-analysis or additional synthesis techniques. However, the diversity of study designs, outcomes, and metrics in the included studies limits the applicability of such methods in this context. For instance, many of the quantitative studies assess different aspects of UHC (e.g., access, affordability, efficiency) using varied measures, making statistical pooling impractical. Instead, the systematic review employs a comprehensive narrative synthesis approach, which aligns well with the review's objective of holistically evaluating the impact of CHIS on UHC. This method allows for an in-depth exploration of themes across studies and provides actionable insights while addressing the inherent heterogeneity in the data. We believe this approach adequately supports the study's conclusions and maintains methodological rigor.

Comment 2: Regional Diversity: Address geographic limitations by including more diverse regional data to generalize findings better.

Authors response: Dear reviewer, thank you for your valuable feedback. We acknowledge the potential geographic limitations of the included studies and it reflects the availability of published research, as few studies specifically address rural or less-studied regions in Saudi Arabia. We have already highlighted this as a limitation in the manuscript and discussed its implications for the generalizability of our findings, see lines 381-382. We hope our acknowledgment of this gap encourages future research to explore healthcare dynamics in underrepresented regions.

Comment 3: Practical Applications: Expand discussion on practical implications for insurance providers and beneficiaries.

Authors response: We have expanded the discussion section to include practical implications for insurance providers and beneficiaries. Specifically, we highlight how providers can use these findings to enhance service delivery and efficiency, while beneficiaries benefit from improved access and affordability. We believe this addition strengthens the manuscript's relevance to both stakeholders and policymakers. The following text was added to the manuscript “The findings of this review have significant practical implications for insurance providers and beneficiaries. For providers, the adoption of standardized benefit packages and monitoring frameworks, as observed under CHIS, highlights opportunities to enhance service efficiency while minimizing out-of-pocket costs. Providers can also address identified gaps, such as disparities in rural access, by investing in infrastructure and tailoring premiums to income levels. For beneficiaries, the expansion of CHIS coverage has the potential to alleviate financial burdens and improve access to essential services, particularly for low-income groups. Policymakers and providers must ensure that reforms prioritize equitable access across socioeconomic and geographic divides to maximize the benefits of CHIS for all stakeholders.” See line 345-353.

Comment 4: Limited coverage of rural healthcare challenges and demographic subgroups.

Authors response: The manuscript acknowledges these limitations in the limitations section but to better address this concern, we have added the following text to emphasize the lack of representation of rural healthcare challenges and demographic subgroups and its implications: "Furthermore, few studies specifically address rural areas, where disparities in healthcare access may be more pronounced, or provide detailed insights into demographic subgroups such as low-income populations, women, or children. This lack of representation limits the ability to generalize findings to these populations, underscoring the need for future research focusing on these groups.". See line 381–385.

Comment 5: Heavy reliance on cross-sectional and self-reported data undermines causal inference.

Authors response: Thank you for this valuable observation. The reliance on cross-sectional and self-reported data reflects the nature of the available studies included in this review. These studies provide valuable insights into CHIS's impact but inherently limit causal interpretations. Therefore, we believe the current text sufficiently addresses this limitation, emphasizing that the findings are a synthesis of the best available evidence while acknowledging the methodological constraints.

Comment 6: The paper is fit for publication with minor revisions. It makes a significant contribution to understanding the role of CHIS in achieving UHC and offers practical insights for policymakers in Saudi Arabia and beyond. Addressing the identified gaps will further enhance its impact and utility.

Authors response: Dear reviewer, thank you for your encouraging feedback and thoughtful suggestions. We have carefully addressed the identified gaps and made revisions to enhance the clarity and impact of the manuscript. We hope our responses to your comments are satisfactory and that the revised manuscript meets your expectations.